bioinformatics/biochemistry/computational biology

SARS-CoV-2, SARS-CoV-2 proteome, T cell epitopes, immunoinformatics, immunopathology, cytokine storm

**Author for correspondence:**
Seema Mishra
e-mail: seema_uoh@yahoo.com

# Designing of cytotoxic and helper T cell epitope map provides insights into the highly contagious nature of the pandemic novel coronavirus SARS-CoV-2

## Seema Mishra

Department of Biochemistry, School of Life Sciences, University of Hyderabad, Hyderabad, India

SM, 0000-0002-4093-7899

Novel coronavirus, SARS-CoV-2, has emerged as one of the deadliest pathogens of this century, creating an unprecedented pandemic. Belonging to the betacoronavirus family, it primarily spreads through human contact via symptomatic and asymptomatic transmission. Despite several attempts since it emerged, there is no known treatment in the form of drugs or vaccines. Hence, work on developing a potential multi-subunit vaccine is the need of the hour. In this study, attempts have been made to find globally conserved epitopes from the entire set of SARS-CoV-2 proteins as there is as yet, no clear information on the immunogenicity of these proteins. Using diverse computational tools, a ranked list of probable immunogenic, promiscuous epitopes generated through all the three main stages of antigen processing and presentation pathways has been prioritized. Moreover, several useful insights were gleaned during these analyses. One of the most important insights is that all of the proteins in this pathogen present unique epitopes, so that the targeting of a few specific viral proteins is not likely to result in an effective immune response in humans. Due to the presence of these unique epitopes in all of the SARS-CoV-2 proteins, stronger immune responses generated by T cell hyperactivation may lead to cytokine storm and immunopathology and consequently, remote chances of human survival. These epitopes, after due validation *in vitro*, may thus need to be presented to the human body in that form of multi-subunit epitope-based vaccine that avoids such immunopathologies.

# 1. Introduction

Novel coronavirus (SARS-CoV-2), also known as 2019-nCoV, first emerged in population in December 2019 and has rapidly gained foothold across the world resulting in WHO declaring it as a pandemic (https://www.who.int/emergencies/diseases/novel-coronavirus-2019). It causes COVID-19 disease with significant mortality rate. As there is currently no known cure, urgent studies are needed in order to push forward drug and vaccine design and development. Recently, about 77 drugs were identified by the world's fastest supercomputer, Summit, against viral spike protein [1]. Immunoinformatics tools have proven crucial time and again in relation to cancer immunotherapy [2,3]. In the absence of effective drugs to date, vaccination is indispensable in order to prevent infections or cure an entire population. As of 15 May 2020, WHO has put forward a draft which identifies eight vaccines in clinical evaluations and 110 candidate vaccines in preclinical evaluations (updated to 26 vaccines in clinical evaluations and 139 candidate vaccines in preclinical evaluations as of 31 July 2020) [4]. More important is the fact that since this COVID-19 disease has affected almost all of the world's population, the vaccine coverage needs to be extensive. In the context of HLA epitope-based multi-subunit vaccine, enlisting promiscuous epitopes binding to a variety of HLA alleles for wider dissemination is crucial. A promiscuous epitope is defined as that epitope which has the capability to bind to multiple HLA alleles. In this regard, *in silico* approaches will be significantly useful in helping develop a preventive approach or a cure in as fast a manner as possible. Vaccines can be administered as prophylactic and even as therapeutic vaccines; as an example, anti-HBV vaccines are currently being developed as therapeutic vaccine candidates. Cytotoxic T cell immune responses have been observed in close relatives, SARS and MERS [5,6], and hence, in SARS-CoV-2 case also, cytotoxic T cell-coordinated immune response along with helper T cell response is crucial. Based on the newly available SARS-CoV-2 genome sequence, this study has been embarked upon with the clear objective of providing a ranked list of highly probable and effective promiscuous epitopes with no human cross-reactivity. Interestingly, several useful insights into the deadly nature of this pathogen were also gleaned along the way.

SARS-CoV-2 genome submitted by CDC, Atlanta (GenBank accession number: MT106054.1 submitted on 24 February 2020) is 29 882 bp in length. Being 100% identical to the reference sequence NC_045512.2 from Wuhan, China, it harbours multiple structural, non-structural and accessory proteins essential or playing a role at various stages of the viral life cycle. This SARS-CoV-2 genome is found 82.3% identical to SARS-CoV genome (NC_004718.3), using NCBI BLASTn tool. In brief, the sequence of proteins in its RNA genome as per this GenBank accession information (figure 1) is as follows: 5′-ORF1ab-S (Spike/Surface)-ORF3a–E (Envelope)-M (Membrane)-ORF6-ORF7a-ORF8-N (Nucleocapsid)-ORF10-polyA tail-3′, which are usually seen in betacoronaviruses [8].

While the structural proteins, S, E, M and N, are key proteins, several proteins such as ORF3a, ORF7a and ORF8 function as accessory proteins playing a role in the viral pathogenesis. ORF1ab, a polyprotein, encodes several non-structural proteins, 15 in number, identified in this genome sequence annotation, including RNA-dependent RNA polymerase (RdRP). The role of structural proteins is determined from their homology to SARS-CoV as well as a few experiments [9]. The expression, localization and function of some SARS-CoV-2 accessory proteins is as yet unclear, although several such proteins have been characterized in SARS-CoV [10], and the roles may be similar in the two viruses. Recently, Gordon *et al*. [11] have cloned and expressed several of these proteins including S, E, M, N, ORF1ab non-structural proteins and accessory proteins ORF3a, ORF6, ORF7a, ORF8 and ORF10. Sequencing studies suggested that the most abundant transcript was N RNA followed by S, ORF7a, ORF3a, ORF8, M, E, ORF6 and ORF7b [7]; ORF7b is identified in this paper [7]). As understood from WHO draft, candidate subunit vaccines are almost all based on spike proteins and very few ones are based on M and N proteins. In view of the scarcity of data on the relevance, immunogenicity and potency/effectiveness of these proteins, any one or more of these proteins may act as prime vaccine candidates. Hence, all of these proteins were used for T cell epitope prediction for the purpose of peptide-based multi-subunit vaccine design and further analyses. The fact that this approach may be better also arises from the previous studies on related SARS-CoV virus [12], wherein more than 50% of the patients had T cell responses against at least one of the two proteins tested, and 25% showed responses against both proteins.

The advantages of epitope-based subunit vaccines as opposed to DNA and live attenuated virus vaccines is that these do not contain live components and so are considered safe. Moreover, these present an antigen or a set of antigens to the immune system with lower risk of side effects [13]. These are also applicable to those people with weakened immune response, which the old people have, and are, therefore, prime targets in the SARS-CoV-2 infection. While cytotoxic T cell (CD8+) response is the key response to immunodominant antigens in destroying a virus-infected cell, helper T

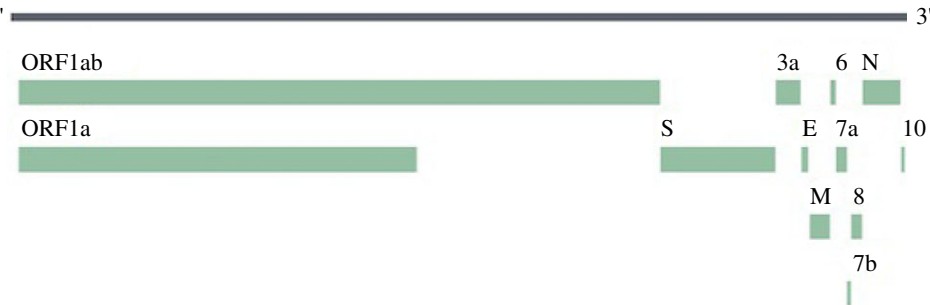

**Figure 1.** Depiction of the location of proteins in the SARS-CoV-2 proteome. Picture taken from NCBI Datasets (https://www.ncbi.nlm.nih.gov/datasets/). ORF7b was identified later in a paper [7].

cells (CD4+) prime and maintain cytotoxic T cells as well as B cells and so, an effective immunotherapeutic product must contain both types of T cell epitopes. These T cell epitopes need to be both high binders to their respective HLA alleles as well as be immunogenic. Further analyses using clustering provided us with consensus epitopes harbouring both CD8+ and CD4+ T cell epitopes, thereby eliminating redundant sequences across target proteins and alleles. These clustered epitopes could elicit stronger cellular immune responses to viral proteins. As opposed to the common perception that membrane and spike proteins may confer better immunogenic ability, an interesting perception is found from this study that it may be the opposite case in the context of SARS-CoV-2 T cell epitopes when studied across populations with different HLA-I supertypes. It should be noted that antibody responses may preferentially target membrane and spike proteins, given their locations on the virus surface, and that this current analysis is geared towards T cell epitopes.

# 2. Results and discussion

## 2.1. Promiscuous, immunogenic Cytotoxic T lymphocyte (CTL) epitopes

To predict potential CTL epitopes against whole coronavirus proteome, predicted proteins or otherwise, CTL epitope prediction was done using PickPocket 1.1 and NetCTLpan 1.1, using the same HLA supertypes. In total, 12 representative supertypes present by default in both the tools were taken. These supertypes are present across populations, and hence, are a representative of the entire world. Further, these two prediction algorithms were used to predict and generate a consensus list of top high binders and promiscuous epitopes across several proteins and supertypes. The consensus list was chosen to increase the prediction accuracy from the two different algorithms. While NetCTLpan uses neural network algorithm, PickPocket works on the basis of position-specific weight matrices. NetCTLpan, in addition to HLA binding, also predicts TAP transporter binding and C-terminal proteasome cleavage predictions. The total number of CTL epitopes generated was 9621 across 10 SARS-CoV-2 proteins including ORF1ab polyprotein. A common list of nine amino acids-long, high binders was generated among topmost epitopes in each protein for each allele, and a total of 122 epitopes were enlisted. These common, promiscuous CTL epitopes are enlisted in tables 1 and 2 as ranked order. It is found that very few promiscuous epitopes could be seen in the case of surface and membrane proteins in topmost epitopes common to both the prediction algorithms [14]. These proteins harbour many potential, unique epitopes across the two prediction tools, leading to the surmise that these two proteins will not be potent, promiscuous immunogens across populations. Nevertheless, a few common promiscuous epitopes across prediction algorithms, although not belonging to the top-ranked ones, were enlisted for these two proteins. One of these epitopes, FVFLVLLPL, the signal peptide in surface/spike protein, has been found to harbour a mutation, L5F, in many strains of 13 countries in distinct phylogenetic clades and L8 V/W mutation is present in Hong Kong [15]. These authors further suggest that L5F mutation might be a sequencing artefact, or may be due to recurrent homoplasy. Epitopes belonging to the spike protein enlisted here do not harbour this residue in the sequences, the D614G mutation, said to be the dominant form in variants in Europe and India. The highest number of common top-ranking epitopes is seen in the case of nsp7 of ORF1ab followed by ORF10, ORF8, ORF6 and ORF3a proteins. Among structural proteins, envelope protein provided the highest number of such epitopes. Venn diagram analysis depicted no

**Table 1.** Top-ranked sequences of CTL epitopes common across HLA supertypes (HLA-A*01:01, HLA-A*02:01, HLA-A*03:01, HLA-A*24:02, HLA-A*26:01, HLA-B*07:02, HLA-B*08:01, HLA-B*27:05, HLA-B*39:01, HLA-B*40:01, HLA-B*58:01, HLA-B*15:01) and across the two prediction algorithms used for nine SARS-CoV-2 proteins.

| epitope sequences | | | | | | | | |
| spike (surface)[a] | ORF3a | envelope | membrane[a] | ORF6 | ORF7a | ORF8 | nucleocapsid | ORF10 |
| --- | --- | --- | --- | --- | --- | --- | --- | --- |
| FVFLVLLPL | FTSDYYQLY | LTALRLCAY | FLFLTWICL | KVSIWNLDY | GTYEGNSPF | IQYIDIGNY | FAPSASAFF | YINVFAFPF |
| YLQPRTFLL | FLYLYALVY | LIVNSVLLF | YFIASFRLF | NLDYIINLI | FLIVAAIVF | GIIITVAAF | NTASWFTAL | MGYINVFAF |
| | FVTVYSHLL | FLAFVVFLL | | HLVDFQVTI | ITLATCELY | SFYEDFLEY | LQLPQGTTL | VFAFPFTIY |
| | LLYDANYFL | IVNSVLLFL | | MFHLVDFQV | ILFLALITL | SLVVRCSFY | | IAQVDVVNF |
| | FVCNLLLLF | LVKPSFYVY | | VTIAEILLI | CVRGTTVLL | QSCTQHQPY | | AFPFTIYSL |
| | YYQLYSTQL | TLAILTALR | | ILLIMRTF | KIILFLALI | YVVDDPCPI | | FPFTIYSLL |
| | WLIVGVALL | | | WNLDYIINL | ELYSPIFLI | TVSCSPFTI | | NSRNYIAQV |
| | YLYALVYFL | | | TIAEILLII | | EYHDVRVVL | | RMNSRNYIA |
| | | | | SIWNLDYII | | HFYSKWYIR | | NVFAFPFTI |
| | | | | | | | | QVDVVNFNL |
| | | | | | | | | FTIYSLLLC |

[a]Not among top-ranked ones in NetCTLpan results.

Table 2. Top-ranked sequences of CTL epitopes common across HLA supertypes (HLA-A*01:01, HLA-A*02:01, HLA-A*03:01, HLA-A*24:02, HLA-A*26:01, HLA-B*07:02, HLA-B*08:01, HLA-B*27:05, HLA-B*39:01, HLA-B*40:01, HLA-B*58:01, HLA-B*15:01) and across the two prediction algorithms used, for SARS-CoV-2 ORF1ab polyprotein.

| leader protein (nsp1) | nsp2 | nsp3 | nsp4 | 3C-like proteinase | nsp6 | nsp7 | nsp8 | nsp9 | nsp10 | RdRP | helicase | 3′-5′ exonuclease | endoRNAse | ribose methyl transferase |
|---|---|---|---|---|---|---|---|---|---|---|---|---|---|---|
| HVGEIPVAY | NMMVTRNNTF | LWAEWFLAY not among top scorers in many alleles | VVWAHNTLLF all alleles except HLA-A*02:01, HLA-A*03:01, HLA-B*07:02, HLA-B*08:01, HLA-B*27:05, HLA-B*39:01, HLA-B*40:01 | QTFSVLACY | MLLYCELGY all alleles except HLA-A*02:01, HLA-A*24:02, HLA-*07:02, HLA-B*39:01, HLA-B*58:01 | MKVSLLSVLL | SSLPSYAAF | CTDDNALAY | FAVDAAKAY | MVMCGGSLY among top scoring in all alleles, except HLA-A*02:01, HLA-A*24:02, HLA-B*08:01, HLA-B*39:01 | YVFCTVNAL all alleles except HLA-A*01:01, HLA-A*03:01, HLA-A*24:02, HLA-B*58:01 | HSIGFDVVY present in HLA-A*01:01, HLA-A*03:01, HLA-A*26:01, HLA-B*07:02, HLA-B*27:05, HLA-B*40:01, HLA-B*58:01, HLA-B*15:01; | TTLPVNVAF | YVMHANYIF all alleles except HLA-A*03:01, HLA-B*27:05 |
| VMVELVAEL | | | | RTILGSALL | LIISVTSNY all alleles except HLA-A*02:01, HLA-A*24:02, HLA-B*08:01, HLA-B*39:01 | SLLSVLLSM | ISMDNSPNL | KSDGTGTIY | STVLSFCAF | TIMADLVYAL among top scoring in all alleles, except HLA-A*01:01, HLA-A*03:01, HLA-A*24:02, HLA-B*08:01, HLA-B*27:05, HLA-B*58:01, | FAIGLALYY top scorer in few alleles in HLA-A*01:01, HLA-A*26:01, HLA-B*58:01, HLA-B*15:01 | | VSININTVY all alleles except HLA-A*02:01 | YSLFDMSKF alleles except HLA-A*02:01, HLA-B*08:01, HLA-B*39:01 |
| HVQLSLPVL | | | | VSFCYMHHM | FLARGIVFM all alleles except HLA-A*01:01, HLA-A*03:01, HLA-A*24:02, HLA-B*27:05, HLA-B*40:01, HLA-B*58:01, | KMVSILSVL | AMQTMLFTM | GTGTIYTEL | PANSTVLSF | LMIERFVSL all alleles except A*01:01, A*03:01, A*26:01, HLA-B*58:01 | | | LLLDDFVBI all alleles except HLA-A*03:01 and HLA-B*27:05 | |

**Table 2.** (*Continued.*)

| leader protein (nsp1) | nsp2 | nsp3 | nsp4 | 3C-like proteinase | nsp6 | nsp7 | nsp8 | nsp9 | nsp10 | RdRP | helicase | 3′-5′ exonuclease | endoRNAse | ribose methyl transferase |
|---|---|---|---|---|---|---|---|---|---|---|---|---|---|---|
| HLIDGTCGL | | | | FLNRFTTTL | | CTSVVLLSV | TTFTYASAL | TELEPPCRF | FGGASCCLY | | | | SQLGGLHLL all alleles except HLA-A*01:01, HLA-A*03:01, HLA-A*26:01, HLA-B*07:02, | |
| POLEQPYVF | | | | HSMQNCVLK all alleles except HLA-A*01:01, HLA-A*02:01, HLA-A*24:02, HLA-A*26:01, HLA-B*07:02, HLA-B*08:01, HLA-B*27:05, HLA-B*39:01, HLA-B*40:01, HLA-B*58:01, HLA-B*15:01 | | TSVVLSVL | | YFIKGLNNL | ITVTPEANM | | | | | |
| QLEQPYVFI RTAPHGHVM | | | | | | KLWAQCVQL VLLSVLQQL EMLDNRATL EAFEKMVSL DVKCTSVIL LAKDTTEAF LHNDILLAK LSMQGAVDI VQLHNDILL | | ALAYYNTTK GMVLGSLAA FVLALLSDL in some alleles, after around top 30 | VLSFCAFAV YLASGGQPI | | | | | |

common epitopes at all across proteins and alleles. Even though SARS-CoV-2 RBD (331–527) is shown to harbour epitopes for eliciting neutralizing antibodies [16,17], this region is not present in the enlisted data for CTL epitopes. However, receptor-binding motif (RBM) region (437–508), the ACE-2 binding motif of this RBD provided immunogenic HTL epitopes, which are detailed below in the section on promiscuous HTL epitopes. Immunogenicity prediction of these proteins (table 3) showed that 71 of these 122 epitopes had a positive immunogenicity score. A clear correlation between HLA binding and immunogenicity in terms of high scores is seen in many of these cases, lending support to the theory that these selected epitopes may mount a high immune response *in vitro* and *in vivo*, too. Further, conserved residues between SARS-CoV-2 and other HCoV and MERS species were found from multiple sequence alignments (MSAs) and found in several of these epitopes (electronic supplementary material, figures S1–S9). As the NCBI RefSeq sequence of SARS-CoV was unclear in the proper annotations for respective proteins, it could not be used in MSA studies. It is observed that most of the epitopes with conserved residues belonged to ORF1ab region (table 3), and epitopes belonging to this region may act as vaccine candidates targeting MERS and other HCoV species, in addition to SARS-CoV-2.

During these CTL epitope identification studies, it was also found that many epitopes identical in sequence as SARS-CoV epitopes found previously in spike, membrane, nucleocapsid and ORF3a proteins [18], were in the lower ranking positions, in the case of different alleles, and many were not common across alleles, so confidence could not be gathered in enlisting these. However, in the ORF3a case, one epitope harbouring both CD8+ and CD4+ T cell epitopes, PLQASLPFGWLVIGV, among the three most frequently recognized by T cells [19], was also present among the top-ranked ones in our study (table 1). Purely for the sake of information to the readers, these T cell epitope data recognized in humans/transgenic mouse in the case of SARS-CoV, that are same/similar to lower ranking T cell epitopes in SARS-CoV-2, are provided as electronic supplementary material, table S1.

## 2.2. Promiscuous, immunogenic helper T lymphocyte (HTL) epitopes

All of the 10 SARS-CoV-2 proteins, predicted or otherwise, were also studied for helper T cell epitope generation using a well-validated prediction tool, NetMHCIIpan, in addition to an immunogenicity prediction tool, CD4episcore, which predicts epitopes based on both HLA-binding and immunogenicity. Prominent HLA-II alleles studied using NetMHCIIpan were: HLA DRB1 alleles, specifically, DRB1*01:01, DRB1*03:01, DRB1*07:01, DRB1*09:01, DRB1*10:01, DRB1*11:01 and DRB1*15:01, because these alleles are found to be frequent across populations ranging from North America, India, Japan, China, Africa and Europe (allelefrequencies.net). The alleles present in CD4episcore are: HLA-DRB1:03:01, HLA-DRB1:07:01, HLA-DRB1:15:01, HLA-DRB3:01:01, HLA-DRB3:02:02, HLA-DRB4:01:01 and HLA-DRB5:01:01.

Helper T lymphocyte epitopes are typically 15 amino acid residues long. High throughput data for these epitopes was analysed manually to identify common epitopes across alleles and 10 coronaviral proteins.

From NetMHCIIpan studies, a total of 1802 promiscuous HTL epitopes (same epitope is predicted to be bound to multiple alleles) selected till rank 2% which are strong binders (or till rank 10%, weak binders in the case that strong binders were not found) were generated. Among these epitopes, 649 epitopes (15-mer) were found to be immunogenic by CD4episcore across all alleles. Another immunogenicity prediction tool, ITcell, was used to predict immunogenic epitopes across only two alleles, DRB1*01:01 and DRB1*15:01, as it uses PDB files for TCR which are available for these two DRB1 alleles, and there was no TCR structure in PDB for other HLA class-II alleles studied. Also, ITcell predicts 12-mer HTL epitopes. Taking ITcell results into account, top-scoring common immunogenic epitopes to both these immunogenicity prediction tools were 95 in number and were taken for further analysis. These also included some of the epitopes binding to the other HLA-DRB1 alleles studied. This can be explained on the basis of observations that among all HLA-II molecules, there exists a high degree of repertoire overlap, reflecting multiple binding partners. This is most probably due to the backbone interactions rather than anchor residues playing a major role [20]. Among these, top 50 high-scoring immunogenic candidate epitopes are tabulated in table 4. A complete list of these and other epitope candidates are provided in electronic supplementary material, table S2. This list also provides immunogenic HTL epitopes in RBM region (437–508), the ACE-2 binding motif of RBD of surface protein, which has been demonstrated to elicit neutralizing antibodies [16]. The whole dataset of HLA-I and HLA-II epitopes across these mentioned and several other HLA-II alleles is available as supplementary information (electronic supplementary material, tables S4–S6).

**Table 3.** Immunogenic CTL epitopes across proteins, sorted by high HLA-I binding, high immunogenicity and conservation of residues in multiple sequence alignment (MSA); epitopes in red font are those nonameric CTL epitopes either existing as a part of longer epitopes binding to HLA-II alleles, or these are clustered together as longer sequences, blue highlights depict sequences showing the presence of conserved residues.

| Epitope | Protein | Peptide start | Peptide end | Immunogenicity score | HLA-I epitopes clustered with HLA-II epitopes | Residue conservation in MSA |
|---|---|---|---|---|---|---|
| FLFLTWICL | Membrane | 26 | 34 | 0.35397 | Singleton | F26, L27, F28 semi-conserved |
| VFAFPFTIY | ORF10 | 6 | 14 | 0.34042 | In 10-membered group | No conservation |
| GIIITVAAF | ORF8 | 8 | 16 | 0.30966 | In 5-membered group | I9,I10,V13, F16 semi conserved |
| IQYIDIGNY | ORF8 | 71 | 79 | 0.30242 | Singleton | I71, Q72 fully-conserved; I74 semi-conserved |
| NVFAFPFTI | ORF10 | 5 | 13 | 0.30241 | In 10-membered group | No conservation |
| FLAFVVFLL | Envelope | 20 | 28 | 0.30188 | Singleton | V25, L28 semiconserved |
| TIAEILLII | ORF6 | 10 | 18 | 0.30101 | In 23-mem bered group | No conservation |
| FLIVAAIVF | ORF7a | 101 | 109 | 0.29611 | Singleton | No conservation |
| KVSIWNLDY | ORF6 | 23 | 31 | 0.29343 | In 23-membered group | No conservation |
| VTIAEILLI | ORF6 | 9 | 17 | 0.28951 | In 23-membered group | No conservation |
| MGYINVFAF | ORF10 | 1 | 9 | 0.28694 | In 10-membered group | No conservation |
| YINVFAFPF | ORF10 | 3 | 11 | 0.28259 | In 10-membered group | No conservation |
| SFYEDFLEY | ORF8 | 103 | 111 | 0.28049 | Singleton | L109 fully-conserved; S103, F104, E106, D107 semi-conserved |
| WNLDYIINL | ORF6 | 27 | 35 | 0.24894 | In 23-membered group | No conservation |
| NLDYIINLI | ORF6 | 28 | 36 | 0.24642 | In 23-membered group | No conservation |
| NTASWFTAL | Nucleocapsid | 48 | 56 | 0.22775 | Singleton | S51 fully-conserved, F53, L56 semiconserved |
| TLAILTALR | Envelope | 30 | 38 | 0.1989 | In 8-membered group | L31, A32, I33, L34, R38 semi-conserved |
| ILFLALITL | ORF7a | 4 | 12 | 0.1895 | In 6-membered group | No conservation |
| WLIVGVALL | ORF3a | 45 | 53 | 0.18314 | Singleton | I47, V48 and L53 semi conserved |
| EYHDVRVVL | ORF8 | 110 | 118 | 0.1807 | Singleton | No conservation |
| QVDVVNFNL | ORF10 | 29 | 37 | 0.17787 | In 3-membered group | No conservation |
| AFPFTIYSL | ORF10 | 8 | 16 | 0.1775 | In 10-membered group | No conservation |
| KIILFLALI | ORF7a | 2 | 10 | 0.16214 | In 6-membered group | No conservation |
| ILLIIMRTF | ORF6 | 14 | 22 | 0.16098 | In 23-membered group | No conservation |
| CVRGTTVLL | ORF7a | 23 | 31 | 0.1536 | Singleton | No conservation |
| SIWNLDYII | ORF6 | 25 | 33 | 0.15011 | In 23-membered group | No conservation |
| YLYALVYFL | ORF3a | 107 | 115 | 0.13151 | In 7-membered group | No conservation |
| YLQPRTFLL | Surface/spike | 269 | 277 | 0.1305 | Singleton | L270 fully conserved, L276, L277 semi conserved |
| LLYDANYFL | ORF3a | 139 | 147 | 0.11841 | In 7-membered group | No conservation |
| ITLATCELY | ORF7a | 23 | 31 | 0.10084 | Singleton | No conservation |
| HLVDFQVTI | ORF6 | 3 | 11 | 0.0982 | In 6-membered group | No conservation |
| NSRNYIAQV | ORF10 | 22 | 30 | 0.09731 | In 9-membered group | No conservation |
| IAQVDVVNF | ORF10 | 27 | 35 | 0.09546 | In 3-membered group | No conservation |
| MFHLVDFQV | ORF6 | 1 | 9 | 0.09154 | In 6-membered group | No conservation |
| YFIASFRLF | Membrane | 95 | 103 | 0.06887 | In 14-membered group | Y95, F96, S99, R101, L102 fully conserved, F103 semi-conserved |
| FPFTIYSLL | ORF10 | 9 | 17 | 0.05708 | In 10-membered group | No conservation |
| FVFLVLLPL | Surface/spike | 2 | 10 | 0.04076 | Singleton | F2, V3, F4, I5, V6 semi conserved |
| ELYSPIFLI | ORF7a | 95 | 103 | 0.03913 | Singleton | No conservation |
| FLYLYALVY | ORF3a | 105 | 113 | 0.03563 | Singleton | No conservation |
| LTALRLCAY | Envelope | 34 | 42 | 0.01886 | In 8-membered group | L39, C40 fully-conserved; L34, R38 semi- conserved |

**ORF1ab**

| Epitope | Protein | From | To | Immunogenicity score | HLA-I epitopes clustered with HLA-II epitopes | Residue conservation in MSA |
|---|---|---|---|---|---|---|
| LVAEWFLAY | nsp3 | 1505 | 1513 | 0.45285 | Singleton | L1505, L1511, A1512 semi conserved, Y1513 fully conserved |
| FLARGIVFM | nsp6 | 184 | 192 | 0.3263 | Singleton | L185, R187 semi-conserved |
| HVGEIPVAY | Leader | 110 | 118 | 0.28861 | Singleton | No conserved residue |
| GTGTIYTEL | nsp9 | 61 | 69 | 0.26744 | Singleton | G63 and I65 semi conserved, E68, L69fully conserved |
| FLNRFTTTL | 3C-like proteinase | 219 | 227 | 0.25596 | Singleton | F219,L220,N221 semi conserved L298, D299, D300, F301, V302 fully conserved; L297, I305 semi-conserved |
| LLLDDFVEI | EndoRNAse | 297 | 305 | 0.24386 | Singleton | |
| LMIERFVSL | RdRp | 854 | 862 | 0.24273 | is part of 6-membered group | E857, R858, V860, S861, L862 fully-conserved, L854, M855, I856, F859 semi-conserved, |
| VMVELVAEL | Leader | 84 | 92 | 0.23373 | Singleton | No conserved residue |
| HSIGFDYVY | 3'-5'exonuclease | 229 | 237 | 0.23318 | Singleton | H229, D234, Y235, Y237 fully-conserved; S230, V236 semi-conserved |
| VSIINNTVY | EndoRNAse | 24 | 32 | 0.22161 | Singleton | S25, I26, N28,N29,T30,V31 semi-conserved |
| KSDGTGTIY | nsp9 | 58 | 66 | 0.22152 | Singleton | S59, D60, G63, I65 semi-conserved |
| VLSFCAFAV | nsp10 | 13 | 21 | 0.17009 | is part of 14-membered group | V13, S15 and A20 semi-conserved, L14, F19 and V21 fully-conserved |
| ITVTPEANM | nsp10 | 55 | 63 | 0.16515 | Singleton | I55 fully-conserved, T56,T58, E60, A61, N62 semi-conserved |
| LHNDILLAK | nsp7 | 35 | 43 | 0.15288 | is part of 20-membered group | H36,N37,I39 fully-conserved, D38 semi-conserved |
| VQLHNDILL | nsp7 | 33 | 41 | 0.14937 | is part of 20-membered group | H36,N37,I39 fully-conserved, D38 semi-conserved |
| VVAFNTLLF | nsp4 | 314 | 322 | 0.1449 | is part of 16-membered group | V315, T319, L320 semi-conserved |
| LAKDTTEAF | nsp7 | 41 | 49 | 0.13402 | is part of 20-membered group | D44, A48 fully |
| EMLDNRATL | nsp7 | 74 | 82 | 0.11684 | Singleton | L82 fully-conserved; E74,D77,A80 semi-conserved |
| RTAPHGHVM | Leader | 77 | 85 | 0.11636 | Singleton | No conserved residue |
| FAIGLALYY | Helicase | 291 | 299 | 0.09181 | is part of 10-membered group | I293, G294, Y299 fully, A292,L295,A296,L297,Y298 semi-conserved |
| TMADLVYAL | RdRp | 123 | 131 | 0.08282 | Singleton | T123, M124, D126, L127, A130, L131 fully-conserved, Y129 semi-conserved |
| YVMHANYIF | Ribose methytransferase | 222 | 230 | 0.0822 | Singleton | H225, A226, N227, Y228, F230 fully-conserved, M224, I229 semi-conserved |
| MLVYCFLGY | nsp6 | 211 | 219 | 0.07782 | Singleton | L212, Y214,G218 fully-conserved, M211, L217, Y219 semi-conserved |
| YVFCTVNAL | Helicase | 355 | 363 | 0.07781 | Singleton | Y355, F357, T359, N361, A362, L363 fully-conserved, V356, V360 semi-conserved |
| TTLPVNVAF | EndoRNAse | 47 | 55 | 0.07705 | Singleton | T47, P50, N52, A54 fully-conserved, T48, V51, V53,F55 semi-conserved |
| SQLGGLHLL | EndoRNAse | 243 | 251 | 0.07388 | Is part of 8-membered group | G246,G247,L248,H249,L250,L251 fully-conserved; L245 semi-conserved |
| CTDDNALAY | nsp9 | 23 | 31 | 0.07355 | Singleton | T24, A30 semi-conserved |
| TELEPPCRF | nsp9 | 67 | 75 | 0.06065 | Singleton | E68, L69, P71, P72, C73, 75 fully-conserved; E70 and R74 semi-conserved |
| ALAYYNTTK | nsp9 | 28 | 36 | 0.05473 | is part of 6-membered group | Y32 fully-conserved, A30, N33 semi-conserved |
| NMMVTNNTF | nsp2 | 625 | 633 | 0.03347 | is part of 6-membered group | F633 semi-conserved |
| QLEQPYVFI | Leader | 63 | 71 | 0.0049 | is part of 2-membered group | Q66, I71 semi-conserved |

**Table 4.** Top 50 immunogenic sequences from CD4episcore and ITcell tools. Red coloured fonts: common to IT cell immunogenicity epitopes sorted by DRB1*0101 score. Blue highlights: common to ITcell immunogenicity epitopes sorted by DRB1*1501 score. Lime green highlights: immunogenic candidates from CD4episcore and common to ITcell and different from Grifoni *et al. Cell Host and Microbe*, 2020 paper with patent; also those in blue highlights that are different from Grifoni *et al.* [21] paper have been mentioned in the text.

| Protein | Protein Number | Protein Description | Peptide | Peptide start | Peptide end | Combined Score |
|---|---|---|---|---|---|---|
| Membrane | 58 | seq47, seq58 | SYFIASFRLFARTRS | 94 | 108 | 22.026 |
| ORF6 | 28 | seq28, seq44 | ILLIIMRTFKVSIWN-Different from Grifoni | 14 | 28 | 22.64144 |
| nsp4 | 106 | seq106 | FYWFFSNYLKRRVVF | 390 | 404 | 23.04084 |
| ORF6 | 22 | seq3, seq22, seq25, se | EILLIIMRTFKVSIW-different from Grifoni | 13 | 27 | 23.3002 |
| nsp4 | 113 | seq113 | KHFYWFFSNYLKRRV | 388 | 402 | 23.52856 |
| Membrane | 46 | seq46, seq59 | YFIASFRLFARTRSM | 95 | 109 | 23.85188 |
| nsp4 | 109 | seq109, seq120 | HFYWFFSNYLKRRVV | 389 | 403 | 24.61324 |
| nsp2 | 103 | seq90, seq103 | QTFFKLVNKFLALCA | 496 | 510 | 24.64816 |
| nsp6 | 64 | seq64 | AMMFVKHKHAFLCLF | 56 | 70 | 24.94928 |
| nsp4 | 115 | seq115 | TKHFYWFFSNYLKRR | 387 | 401 | 24.98624 |
| ORF6 | 27 | seq2, seq27, seq43 | AEILLIIMRTFKVSI-different from Grifoni | 15 | 29 | 25.0036 |
| nsp2 | 91 | seq91, seq111 | VQTFFKLVNKFLALC | 495 | 509 | 25.1072 |
| Membrane | 43 | seq43, seq64 | IASFRLFARTRSMWS | 97 | 111 | 25.22452 |
| Membrane | 42 | seq19, seq33, seq39, s | ASFRLFARTRSMWSF | 98 | 112 | 25.24444 |
| nsp6 | 60 | seq60, seq80 | AFAMMFVKHKHAFLC | 54 | 68 | 25.25948 |
| nsp6 | 81 | seq62, seq81 | FAMMFVKHKHAFLCL | 55 | 69 | 25.52472 |
| nsp6 | 75 | seq58, seq75 | SAFAMMFVKHKHAFL | 53 | 67 | 26.23016 |
| nsp2 | 105 | seq89, seq105 | SVQTFFKLVNKFLAL | 494 | 508 | 26.38856 |
| RdRp | 5 | seq5, seq71, seq100, s | MPNMLRIMASLVLAR-different from Grifoni | 626 | 640 | 26.47384 |
| Membrane | 60 | seq45, seq60 | FIASFRLFARTRSMW | 96 | 110 | 26.53024 |
| nsp2 | 108 | seq95, seq108 | TFFKLVNKFLALCAD | 497 | 511 | 26.67152 |
| RdRp | 113 | seq9, seq75, seq113, s | AMPNMLRIMASLVLA-different from Grifoni | 625 | 639 | 27.38192 |
| RdRp | 53 | seq53, seq135 | LRIMASLVLARKHTT-different from Grifoni | 630 | 644 | 27.69396 |
| RdRp | 20 | seq20, seq180 | RAMPNMLRIMASLVL | 624 | 638 | 28.1144 |
| nsp4 | 125 | seq125 | STKHFYWFFSNYLKR | 386 | 15 | 28.29724 |
| RdRp | 152 | seq1, seq48, seq70, se | PNMLRIMASLVLARK-different from Grifoni | 627 | 641 | 28.29936 |
| Ribose methy | 6 | seq6, seq48 | GRLIIRENNRVVISS | 278 | 292 | 28.37836 |
| RdRp | 120 | seq11, seq49, seq120 | MLRIMASLVLARKHT | 629 | 643 | 28.38828 |
| RdRp | 74 | seq6, seq45, seq74, se | NMLRIMASLVLARKH-different from Grifoni | 628 | 642 | 28.83568 |
| ORF6 | 45 | seq4, seq20, seq29, se | LLIIMRTFKVSIWNL-different from Grifoni | 15 | 29 | 28.88076 |
| ORF6 | 48 | seq17, seq30, seq48 | LIIMRTFKVSIWNLD | 16 | 30 | 29.11184 |
| RdRp | 99 | seq19, seq83, seq99, s | QMNLKYAISAKNRAR | 541 | 555 | 29.1294 |
| nsp8 | 28 | seq28 | VVLKKLKKSLNVAKS | 33 | 47 | 29.31276 |
| Ribose methy | 7 | seq7, seq47 | KGRLIIRENNRVVIS | 277 | 291 | 29.31836 |
| nsp8 | 27 | seq27 | EVVLKKLKKSLNVAK | 32 | 46 | 29.46256 |
| Ribose methy | 8 | seq8 | RLIIRENNRVVISSD | 279 | 293 | 29.7396 |
| nsp2 | 94 | seq94 | ESVQTFFKLVNKFLA | 493 | 507 | 29.9796 |
| RdRp | 107 | seq86, seq107, seq139 | TQMNLKYAISAKNRA | 540 | 554 | 30.01152 |
| Membrane | 68 | seq21, seq34, seq38, s | SFRLFARTRSMWSFN | 99 | 113 | 30.13772 |
| Nucleocapsid | 49 | seq49 | DQIGYYRRATRRIRG | 82 | 96 | 30.45712 |
| RdRp | 87 | seq24, seq87, seq108, | MNLKYAISAKNRART | 542 | 556 | 30.4728 |
| nsp6 | 22 | seq22, seq92 | VLLILMTARTVYDDG-different from Grifoni | 121 | 135 | 30.78116 |
| Exonuclease | 56 | seq15, seq32, seq56 | AYNMMISAGFSLWVY | 497 | 511 | 30.94884 |
| Nucleocapsid | 48 | seq48 | QIGYYRRATRRIRGG | 83 | 97 | 30.9554 |
| Exonuclease | 53 | seq11, seq30, seq53 | DAYNMMISAGFSLWV | 496 | 510 | 31.22164 |
| nsp6 | 91 | seq20, seq84, seq91 | VVLLILMTARTVYDD-different from Grifoni | 120 | 134 | 31.25796 |
| Helicase | 84 | seq84 | CFKMFYKGVITHDVS | 471 | 485 | 31.35264 |
| Exonuclease | 71 | seq3, seq31, seq39, se | DMTYRRLISMMGFKM-different from Grifor | 48 | 62 | 31.65644 |
| Ribose methy | 49 | seq9, seq49 | SKGRLIIRENNRVVI | 276 | 290 | 31.83036 |
| Nucleocapsid | 50 | seq50 | IGYYRRATRRIRGGD | 84 | 98 | 31.83548 |

## 2.3. CTL and HTL epitope distribution across SARS-CoV-2 proteome

Bar diagram for CTL and HTL immunogenic epitope distribution across proteins (figure 2) shows a general trend with the number of epitopes not correlated with the size of the proteins. The smallest predicted protein, ORF10, is found to provide more CTL epitopes in the context of this study than the larger spike protein. Some previous studies have also found this to be true, wherein capsid and matrix proteins in the viruses studied were found to 'pack significantly more epitopes than those expected by their size' [22]. Some proteins such as ORF6, ORF8, ORF10, envelope and membrane do not have immunogenic HTL epitopes that harbour nonameric CTL epitopes, binding to either HLA-DRB1*0101 and HLA-DRB1*1501, and in some cases to none of the two alleles. Also, leader, nsp7, nsp10 and endoRNAse proteins of ORF1ab did not provide common epitopes between the two immunogenicity prediction tools. The highest number of immunogenic HTL epitopes as predicted by CD4episcore was provided by RdRp, followed by nsp3, nsp4, helicase and spike (surface) protein sequences.

Venn diagram depicted a common list of many epitopes from a single protein across alleles (electronic supplementary material, figure S10). A distinct pattern is to be noted; analysis of HTL epitopes belonging to HLA-DRB1*03:01, HLA-DRB1*11:01 and HLA-DRB1*15:01 indicated the lowest number of common epitopes or none at all across most of the proteins, and can be considered outlier epitopes. Envelope protein was unique in the sense that it did not provide either strong or weak binders to HLA-DRB1*03:01 allele, frequent across North America, Europe, India and Africa. ORF10 was also unique

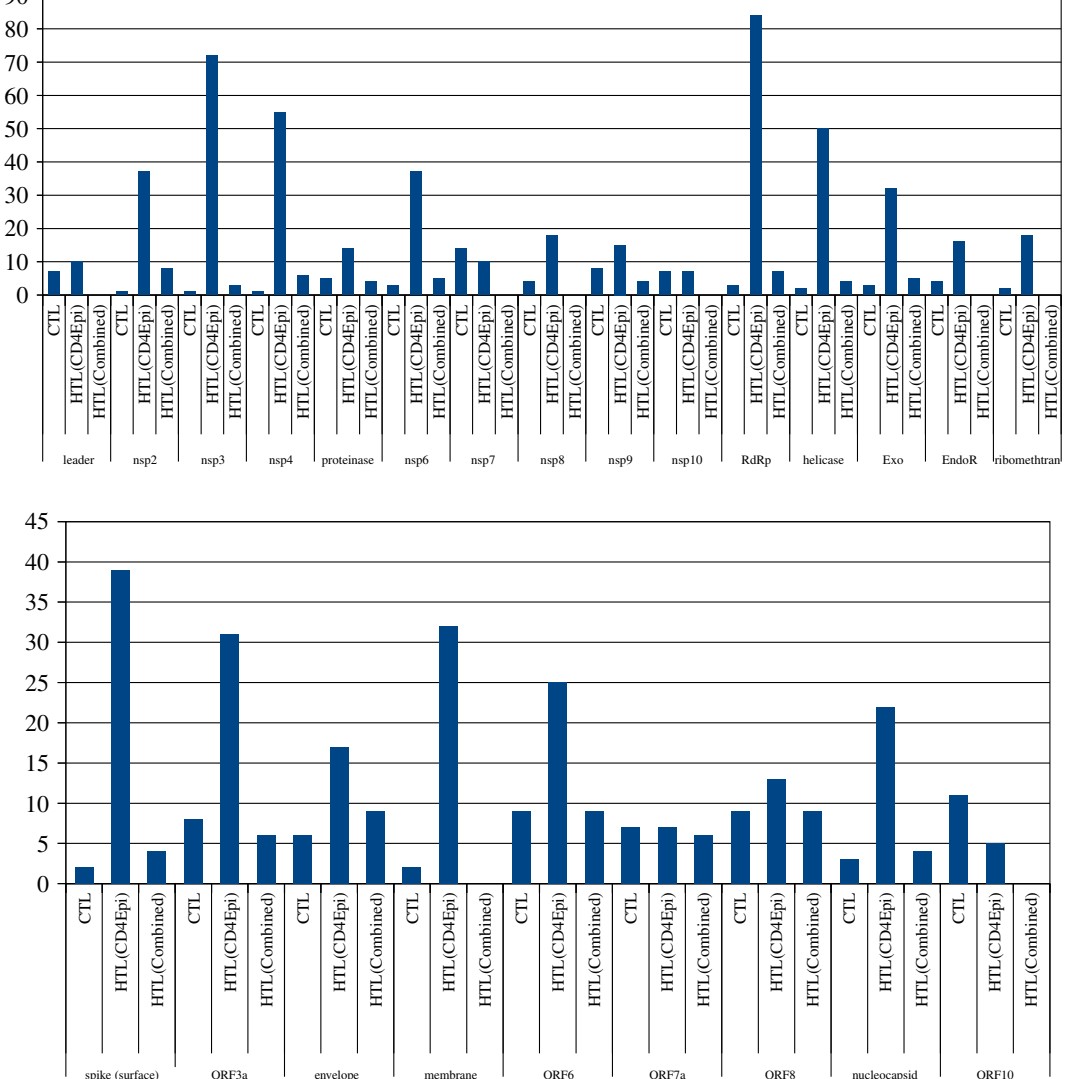

**Figure 2.** Bar diagram for CTL and HTL immunogenic epitope distribution across proteins. HTL(CD4epi) depicts immunogenic epitopes from CD4episcore tool. HTL(Combined) depicts epitopes common to CD4episcore and ITcell tools. First panel: all ORF1ab proteins.

in providing only weak HTL binders to all of the alleles studied. Venn diagram of all these cytotoxic and helper T cell epitopes taken together showed no common epitopes at all across proteins, but within a given protein set, common epitopes could be found. This observation indicates that every protein of SARS-CoV-2 may present antigenic epitopes to the immune system, resulting in a high number of targets. This further lends credence to the theory that multiple T cell epitopes may elicit an immune response in each case, some eliciting strong and some providing weaker responses and therefore, there may be high degree of T cell immunopathology at the infection site. Stronger T cell immune response may cause even the normal, uninfected cells to be attacked while weaker helper T cell immune response, in some protein targets, may cause weak neutralizing antibody responses as well as weak CTL response at varying times during infection. Very recently, one study has pointed to this immune dysregulation [23] in COVID-19 patients with IL6-mediated low HLA-DR expression with sustained cytokine production. Another correspondence paper also pointed to a cytokine storm in context [24]. The involvement of T cells in the development of cytokine storm cannot be ruled out, where preliminary findings show antigen-specific production of IL-6 and TNF-α response in a dead patient's cell culture supernatants and proposed to be carried out further in a larger cohort [25]. Antibody-mediated enhancement of immune response is also not ruled out and can be seen from the fact that all the epitopes present in the list of dominant B cell epitopes (tab. 4 in [21]) belonging to surface, membrane and nucleocapsid protein, are unique, and there may be a higher non-neutralizing antibody level in COVID-19 patients, like in the case of dengue viruses [26].

While this study was at the writing stage, two studies on T cell epitope generation using all proteins [21,27] were published. This present study is different from Grifoni *et al.* [21] study in that two prediction tools with very different algorithms, one using neural network and another using position-specific weight matrices were employed to generate a list of common epitopes, thereby increasing prediction accuracy. Also, Grifoni *et al.* [21] focused mostly on previous SARS coronavirus epitope similarity for predicting epitopes, while this paper identified several novel epitopes across all 10 proteins using two different prediction algorithms in each case. Further, this epitope list comprises common top-scoring epitopes with a higher accuracy and is restricted to highly frequent HLA alleles across populations. Also, in view of the several mutations in SARS-CoV-2 genome distinct from SARS-CoV, these epitopes that are not found from similarity to SARS-CoV epitopes, may be potentially more immunogenic. Most of the novel HTL and CTL epitopes in this study, were distinct from the epitopes predicted by Grifoni *et al.* [21], and were found among top 100 immunogenic candidates predicted by CD4episcore as well as those in common to ITcell predictions (electronic supplementary material, table S2). There was no supplementary material on the website or sequence information of the epitopes in the study from Nguyen *et al.* [27]. Further, their work did not take into account TAP transporter binding predictions as well as HLA-II binding studies, while this study took all three stages of MHC processing and presentation pathway: proteasomal cleavage, TAP transporter binding and MHC class I and II binding as well as immunogenicity studies into account for comprehensive predictions.

## 2.4. Clustering analysis

All of the 1924 CTL and HTL topmost epitopes (122 CTL epitopes and 1802 HTL epitopes) across the proteins studied, of which 1096 were non-redundant, unique epitopes, were then clustered using IEDB epitope cluster analysis tool [28] to make further biologically meaningful decisions. Results analysed suggested that many epitopes were clustered around a given consensus sequence (electronic supplementary material, table S3). The total number of clusters (including subclusters) was 244, and 66 epitopes were singletons not present in a cluster.

The larger clusters harbouring consensus sequences were: VDFQVTIAEILLIIMRTFKVSIWNLDY-IINLIIKN (23 members), KLWAQCVQLHNDILLAKDTTEAFEKMVSLLSVLLSM and TQHQPYVVD-DPCPIHFYSKWYIRVGARKSAPLIEL (20 members each). These clusters across proteins and alleles may be considered immunodominant epitopes and tested first among the ranked list of epitopes.

Among immunogenic 122 CTL epitopes from IEDB and 666 HTL epitopes from CD4episcore, again HLVDFQVTIAEILLIIMRTFKVSIWNLDYIINLII topped the list. Further, among the same immunogenic 122 CTL and 95 HTL epitopes common to two prediction algorithms, CD4episcore and ITcell, VTIAEILLIIMRTFKVSIWNLDYIINL belonging to ORF6 again topped. Moreover, PIHFYSKWYIRV-GARKSAPLIEL belonging to ORF8 and MGYINVFAFPFTIYSLL belonging to ORF10 were also among the top three clustered sequences. It is of interest to note that sequences in the consensus sequence MGYINVFAFPFTIYSLL belonging to ORF10 are weak binders to all the HLA-DRB1 alleles studied, while the nonameric sequences in this consensus sequence are strong binders to all HLA-I supertypes studied.

## 2.5. Cross-reactivity studies

Cross-reactivity analyses against human proteome based on UniProt data (figure 3) indicated that all the immunogenic CTL and HTL epitopes (all HTL epitopes taken from CD4episcore list, removing redundant HTL epitopes; total 719 CTL + HTL epitopes) obtained were not present in human proteome and hence, no cross-reactivity to normal human cells may occur.

## 2.6. ADE and B-cell epitopes

The widespread presence of novel, unique T cell epitopes in the SARS-CoV-2 proteome, is also the main reason that in this paper, B cell epitopes were not studied. Further, including B cell epitopes in the vaccination strategy with T cell epitopes may not be a good strategy, and may even be counter-productive. Even though neutralizing antibody levels are found to be low in COVID-19 patients [29,30], it is expected that CD4 + T cell expansion responses may increase the neutralizing antibody levels [31] and hence, quantifying CD4 + T cell responses using IFN-gamma ELISPOT assays will be useful. This is done in order to minimize the possible immune system backfiring [23,24] due to the presence of too many overlapping as well as non-overlapping epitopes in multi-subunit vaccines. It is suggested that

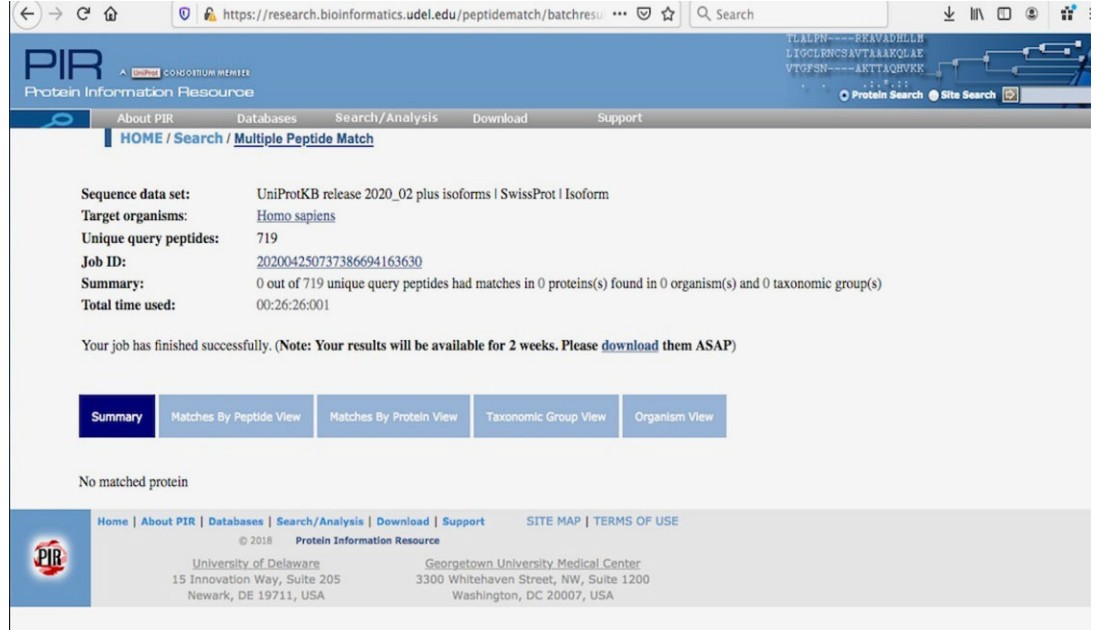

**Figure 3.** Multiple Peptide Match of 719 predicted SARS-CoV-2 coronaviral epitopes against *Homo sapiens* proteome from UniProt.

helper T cell epitopes be chosen so as to elicit an immune response robust enough to prime and maintain neutralizing antibody responses, as well as keep the immunopathology under check. In the proven scenario of immune system backfiring, it may be one possible mechanism by which SARS-CoV-2 may be acting at its deadliest nature. It is indeed, a dangerous pathogen to control, although for effective immunotherapy at a global scale, efforts should already be underway using this ranked list of epitopes. Almost all of its proteins may pose as foreign agents to the human immune system, with each protein contributing several unique, different immunogenic epitopes. This horde of foreign proteins brings down an avalanche of immune system molecules to the infection site, in order to fight the virus. But instead of immune protection, this may lead to immune enhancement or allergic inflammation at the infection site. These analyses demonstrate that coronavirus genome has evolved to be a unique genome. Even as this study is important in pointing out the possible mechanisms such as immunopathologies arising due to T cell hyperactivation, contributing to the contagious nature of SARS-CoV-2, more evidence is required in the form of *in vitro* and *in vivo* experiments.

While many of the proteins studied are found to be expressed and also their functions known by virtue of homology with SARS-CoV, many of the novel ORFs including ORF8 and ORF10 need to be experimentally tested for their functional validation. Experimental MHC-peptide binding and T cell stimulation assays are now required for *in vitro* testing for further refinement and development as potent immunogens to be incorporated as components of multi-subunit vaccines.

## 3. Conclusion

Utilizing all 10 of the SARS-CoV-2 proteins, predicted or otherwise, a ranked list of CTL and HTL epitopes with high HLA-binding affinity, high TAP transport efficiency and high C-terminal proteasomal cleavage ranking has been generated. Incorporating the alleles predominant in the whole world population, two different prediction algorithms were implemented in the identification of common epitopes for creating consensus. Immunogenicity scores for these epitopes have also been predicted in order to further narrow down the list to a few key epitopes that can be experimentally tested. Peptide matching with the human proteome showed no indication of possible cross-reactivity. These epitopes are provided to the scientific community for further development using *in vitro* and *in vivo* assays and saving their time and costs involved in our urgent bid to tackle SARS-CoV-2 infections and ensuing death. This essential list of highly probable epitopes opens up avenues for developing prophylactic and therapeutic interventions and for further understanding of the human immune system responses to this virus.

# 4. Material and methods

## 4.1. Genome sequence

The genome sequence of novel coronavirus was retrieved from GenBank accession number MT106054.1/ RefSeq sequence number NC_045512.2 and the corresponding proteins were retrieved. RefSeq sequences of all of the proteins present in this genomic sequence, ORF10 protein (YP_009725255.1), nucleocapsid phosphoprotein (YP_009724397.2), ORF8 protein (GenBank: QID21074.1; no RefSeq sequence is identified for ORF8), ORF7a protein (YP_009724395.1), ORF6 protein (YP_009724394.1), membrane glycoprotein (YP_009724393.1), envelope protein (YP_009724392.1), ORF3a protein (YP_009724391.1), surface glycoprotein (YP_009724390.1) and ORF1ab (YP_009724389.1) were analysed in order to cover the entire genome of SARS-CoV-2 in view of absence of data on its virulent proteins. Within ORF1ab (full protein accession number: YP_009724389.1), the accession numbers of the following proteins taken were as follows: leader protein—YP_009725297.1, nsp2—YP_009725298.1, nsp3—YP_009725299.1, nsp4—YP_009725300.1, 3C-like proteinase—YP_009725301.1, nsp6—YP_009725302.1, nsp7—YP_009725303.1, nsp8—YP_009725304.1, nsp9—YP_009725305.1, nsp10—YP_009725306.1, RNA-dependent RNA polymerase—YP_009725307.1, helicase—YP_009725308.1, 3′-to-5′ exonuclease—YP_009725309.1, endoRNAse—YP_009725310.1 and 2′-O-ribose methyltransferase—YP_009725311.1. Fasta sequences of all of these proteins were taken as inputs in several T cell epitope prediction and analysis tools.

## 4.2. Cytotoxic T lymphocyte epitope prediction

NetCTLpan v. 1.1 (http://www.cbs.dtu.dk/services/NetCTLpan/, [32]) and PickPocket v. 1.1 (http:// www.cbs.dtu.dk/services/PickPocket/, [33]) were used. All of the parameters used were default parameters. Nonameric peptide epitopes were selected. Epitopes from NetCTLpan were ranked according to the combined score using all three different methods representing antigen processing and presentation steps, and epitopes from PickPocket algorithm were sorted by affinity ($IC_{50}$ values in nM). High-scoring epitopes were chosen as follows: among top 10 in PickPocket and same epitopes among high-scoring ones in NetCTLpan, so as to find common epitopes to increase accuracy. A total of 12 HLA supertypes present in both algorithms were as follows: HLA-A*01:01, HLA-A*02:01, HLA-A*03:01, HLA-A*24:02, HLA-A*26:01, HLA-B*07:02, HLA-B*08:01, HLA-B*27:05, HLA-B*39:01, HLA-B*40:01, HLA-B*58:01 and HLA-B*15:01 [14]. For ORF1ab proteins, because common epitopes could not be found from top scorers in NetCTLpan and PickPocket methods, top 30 candidates were used to select promiscuous epitopes.

## 4.3. Helper T lymphocyte epitope prediction

NetMHCIIpan v. 3.2 (http://www.cbs.dtu.dk/services/NetMHCIIpan/, [34]) was used to predict helper T cell epitopes across several HLA-DRB1 alleles, specifically, DRB1*01:01, DRB1*03:01, DRB1*07:01, DRB1*09:01, DRB1*10:01, DRB1*11:01 and DRB1*15:01. It works on the basis of quantitative MHC-peptide binding affinity data obtained from the Immune Epitope Database (IEDB). A consensus list of 15 amino acids-long ranked epitopes was generated. For generating top-ranked epitopes, these were sorted using descending order of per cent rank. Per cent rank is a normalized prediction score, comparing to the prediction of a set of random peptides [32]. The epitopes with per cent rank less than 2% and less than 10% were considered strong and weak binders, respectively.

## 4.4. Immunogenicity prediction

Immunogenicity is a characteristic property of peptide epitopes that can elicit an immune response. High binding affinity to HLA alleles is not a sufficient criterion for high immunogenicity. Therefore, all the epitopes that were generated as a consensus were checked for their immunogenicity. IEDB immunogenicity tool (http://tools.iedb.org/immunogenicity/, [35]) was used to generate a list of immunogenic CTL epitopes. Immunogenicity of a peptide–MHC complex is predicted based on the physico-chemical properties of amino acids and their positions in the predicted peptide. Specifically, amino acids with large and aromatic side chains and positions 4–6 are more important to the immunogenicity of the peptide being presented. The ranking was done after sorting from higher to lower immunogenicity score [35]. For helper T cell epitopes immunogenicity prediction, CD4episcore [36] and ITcell [37] were used. CD4episcore was developed using neural networks and combines HLA

binding and immunogenicity prediction and outputs a list of immunogenic peptides using a combined score. The authors combined immunogenicity and HLA-binding scores, using the median percentile rank score (HLA_score) of the 7-allele method (ranging from 0 to 100) and combined it with their neural network-based immunogenicity score. This combined score is calculated as follows:

Combined score: (alpha * Imm score) + ((1 − alpha) * HLA_score), where alpha is optimized to 0.4.

The 7 alleles used are: 'HLA-DRB1:03:01', 'HLA-DRB1:07:01', 'HLA-DRB1:15:01', 'HLA-DRB3:01:01', 'HLA-DRB3:02:02', 'HLA-DRB4:01:01' and 'HLA-DRB5:01:01'. The whole HTL epitope sequence list belonging to each protein was given as an input, and IEDB-recommended combined method was selected for scoring. Lower combined scores imply higher immunogenicity according to the authors developing this prediction tool. The immunogenic versus non-immunogenic epitopes cut-off was a combined score of 50 as per CD4episcore paper.

ITcell works on the basis of three stages of MHC-II processing and presentation pathway. These three stages are, in the authors' [37] own words: '….antigen cleavage, MHCII presentation and TCR recognition. First, antigen cleavage sites are predicted based on the cleavage profiles of cathepsins S, B and H. Second, for each 12-mer peptide in the antigen sequence we predict whether it will bind to a given MHCII, based on the scores of modelled peptide-MHCII complexes. Third, we predict whether or not any of the top-scoring peptide-MHCII complexes can bind to a given TCR, based on the scores of modelled ternary peptide-MHCII-TCR complexes and the distribution of predicted cleavage sites.' The scores are given as normalized Z-scores with negative scores implying higher immunogenicity. The epitope sequences as well as PDB files for TCR molecules corresponding to their cognate MHC alleles were given as an input. The PDB ID for files for HLA-DRB1*01:01 and HLA-DRB1*15:01 alleles are 1FYT.pdb and 1YMM.pdb, respectively. PDB files for all other alleles were not available.

## 4.5. Clustering

As globally conserved epitopes are relevant at this time to contain and treat coronavirus infection, the clustering approach was used to find patterns among disparate datasets. In order to group epitopes into several clusters, IEDB epitope cluster analysis tool [28] was applied. All the topmost CTL and HTL epitopes across proteins targets were used as inputs with minimum sequence identity threshold as 70%. Cluster-break algorithm was applied to generate a clear representative sequence.

## 4.6. Cross-reactivity analysis

All the immunogenic CTL and HTL epitopes obtained were used to search against human proteome data from UniProt database (2020_02 release, 181 292 975 sequences as of date 6 May 2020) for any matches to human proteome, thus avoiding cross-reactivity. For this, Multiple Peptide Match tool (https://research.bioinformatics.udel.edu/peptidematch/batchpeptidematch.jsp) of Protein Information Resource was used.

## 4.7. Multiple sequence alignment

MUSCLE (https://www.ebi.ac.uk/Tools/msa/muscle/) tool was used to generate MSAs of all SARS-CoV-2 proteins with corresponding proteins in other HCoV and MERS species. The species chosen and their GenBank accession IDs were: Alpha-CoV: HCoV-NL63 (NC_005831.2), HCoV-229E (NC_002645.1); Beta-CoV: HCoV-OC43 (NC_006213.1), HCoV-HKU1 (NC_006577.2), MERS CoV (NC_019843.3) and SARS-CoV-2 (SARS-CoV-2, accession IDs same as above). The spike protein sequence for SARS-CoV was taken from UniProt (P59594). In view of different/unclear annotations, it was difficult to get corresponding protein sequences from SARS-CoV (RefSeq accession ID NC_004718.3). There are no human CoVs in gamma/delta CoV categories. In addition, bat coronavirus RaTG13 sequences (MN996532.1) were also used.

# Notes

This manuscript has been released as two pre-prints at ChemRxiv, (Mishra, 2020) with one part of the manuscript published with doi:10.26434/chemrxiv.12029523.v2 [14] and another part with doi:10.26434/chemrxiv.12253463.v1 [38]. All the research work was done during lockdown working from home and the author is indebted to Computational Biology field in helping urgently design vaccine for the virus causing the pandemic. The research, must go on!

Data accessibility. Accession IDs of all data taken from publicly available databases analysed in this study are included in this article.

Authors' contributions. S.M. conception or design of the work; acquisition, analysis and interpretation of data; drafted the work, gave final approval for publication.

Competing interests. I declare I have no competing interests

Funding. I received no funding for this study.

Acknowledgements. This author acknowledges the tireless help of researchers working towards understanding SARS-CoV-2 biology and submitting data to GenBank, without which these analyses using Immunoinformatics for probable vaccine candidates would not have been possible. The author further acknowledges the technical help rendered by Aanchal Mishra.

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
