## [Reviewer comments · Royal Society Open Science]

Review History

RSOS-201141.R0 (Original submission)

Review form: Reviewer 1

Is the manuscript scientifically sound in its present form?

Yes

Are the interpretations and conclusions justified by the results?

No

Is the language acceptable?

No

Do you have any ethical concerns with this paper?

No

Have you any concerns about statistical analyses in this paper?

No

Recommendation?

Major revision is needed (please make suggestions in comments)

Comments to the Author(s)

In this work, Mishra explore the immunogenicity of thee produced proteins in the SARS-CoV2 RNA genome. Using 2 different computational approaches that takes into account the stages of antigen processing and presentation, MHC I and MHCII binding, and immunogenicity, the authors proposes a list of peptides in almost all of the predicted SARS-CoV2 proteins that may be tested for possible vaccine candidates. The work is very comprehensive, timely and will be of interest to the field. However, in some places, the data is overinterpreted, and there are a number of awkward statements, and some concerns about the presentation of the work as detailed below:

1. The manuscript would be enhanced with a figure depicting the viral proteins being evaluated.
2. On page 1, line 44, the author suggests that this work would be used to determine virulence of the proteins. The word virulence should be replaced with immunogenicity.
3. The statement on page 1, line 54 in the sentence "...if one protein is attacked by the immune system, weakening the virus, other proteins can continue to assist in its survival." This statement should be revised to reflect the fact that immune targeting of specific proteins is not likely to result in such a response.
4. On page 2, line 2, the author suggests that the presence of unique epitopes in SARS-Cov2 may result in a "...a stronger immune response generated may lead to cytokine storm and immunopathology". While SARS-CoV2 has been shown to induce cytokine storm, it is not likely that this is due to the number of unique epitopes.
5. On page 3, line 6, the author states "...vaccination is indispensable in order to cure an entire population." Vaccines generally do not cure infections, but prevent infection or symptoms. This statement should be corrected.
6. On page 3, line 48, the author states "As opposed to common perception that membrane and spike proteins confer better immunogenic ability, an interesting perception is found from this study that it may be the opposite case in context of SARS-CoV-2 when studied across populations with different HLA-I supertypes." This statement should be qualified to indicate that antibody responses may preferentially target membrane and spike proteins given their locations on the virus, and that this current analysis is geared towards T cell epitopes.
7. The author refers "promiscuous" (e.g. page 5 line 48). The parameters used to determine promiscuous should be stated.
8. On page 6, line 25, in the sentence "...lending support to the theory that these selected epitopes may mount a high immune response in vitro.", the reference to "in vitro" should be "in vivo".
9. On page 13, line 9, the sentence " Another immunogenicity prediction tool, Tcell, was used to predict immunogenic epitopes across two alleles DRB1*01:01 and DRB1*15:01 as it uses PDB files for TCR and there was no structure for other alleles in PDB." What does the author mean by "...PDB files for TCR and there was no structure for other alleles in PDB".
10. The headings and text in the tables, particularly Table 2, need to better described as it's a bit confusing. Particularly, the heading, "clustering with HLA-II epitopes" and "conservation in MSA".
11. The title of Table 2 is "Immunogenic CTL epitopes..." It is not clear if this represents peptides identified as HLA II binders that also bind to HLA I.
12. The author refers to yellow highlights in Table 3, but there is no such yellow highlighting.
13. There is a section on "B cell epitopes", however, this work does not analyze for B cell epitopes. This section should be removed.
14. On page 19, line 51, the author states that "Including B cell epitopes in the vaccination strategy with T cell epitopes may not be a good strategy, and may even be counter-productive." This statement should be revised to reflect that fact that both B and T cell epitopes are important since antibody responses are also critical to generate virus neutralizing antibodies.
15. On page 20, line 9, the author states "This is so done in order to minimize the possible immune system backfiring (22, 23) due to the presence of too many overlapping as well as non-

overlapping epitopes in multi-subunit vaccines." It is not clear what is meant by "immune system backfiring".

16. On page 20, line 37, the sentence "Even as this study is important in pointing out the possible mechanisms in contagious nature of SARSCoV-2, more evidence is required in the form of experiments." It's not clear how the analysis performed in this work point to possible mechanisms of contagious nature of SARS-COV2. This sentence should be revised.

Decision letter (RSOS-201141.R0)

Dear Dr Mishra,

The editors assigned to your paper ("Designing of cytotoxic and helper T cell epitope map provides insights into the highly contagious nature of the pandemic novel coronavirus SARS-CoV-2") have now received comments from reviewers.

The referee and Associate Editor have responded favourably to publication. However, the reviewer raises a number of substantive points regarding both over-interpretation and the need for additional verification of the statistics applied. It will be important to carefully consider and respond to these points. We would like you to revise your paper in accordance with the referee and Associate Editor suggestions which can be found below (not including confidential reports to the Editor). Please note this decision does not guarantee eventual acceptance.

Please submit a copy of your revised paper before 28-Aug-2020. Please note that the revision deadline will expire at 00.00am on this date. If we do not hear from you within this time then it will be assumed that the paper has been withdrawn. In exceptional circumstances, extensions may be possible if agreed with the Editorial Office in advance. We do not allow multiple rounds of revision so we urge you to make every effort to fully address all of the comments at this stage. If deemed necessary by the Editors, your manuscript will be sent back to one or more of the original reviewers for assessment. If the original reviewers are not available, we may invite new reviewers.

If your study uses humans or animals please include details of the ethical approval received, including the name of the committee that granted approval. For human studies please also detail

whether informed consent was obtained. For field studies on animals please include details of all permissions, licences and/or approvals granted to carry out the fieldwork.

- Data accessibility

If you wish to submit your supporting data or code to Dryad (<http://datadryad.org/>), or modify your current submission to dryad, please use the following link:
<http://datadryad.org/submit?journalID=RSOS&manu=RSOS-201141>

- Competing interests

- Authors' contributions

- Acknowledgements

- Funding statement

Kind regards,

Andrew Dunn

on behalf of Dr John Dalton (Associate Editor) and Steve Brown (Subject Editor)
 openscience@royalsociety.org

Associate Editor's comments (Dr John Dalton):

Associate Editor: 1

Comments to the Author:

The paper received a favourable review. However, major revision and additional verification of the statistics applied is asked by the reviewer. The reviewer also felt that the data in some places has been over-interpreted.

Comments to Author:

Reviewers' Comments to Author:

Reviewer: 1

Comments to the Author(s)

In this work, Mishra explore the immunogenicity of thee produced proteins in the SARS-CoV2 RNA genome. Using 2 different computational approaches that takes into account the stages of antigen processing and presentation, MHC I and MHCII binding, and immunogenicity, the authors proposes a list of peptides in almost all of the predicted SARS-CoV2 proteins that may be tested for possible vaccine candidates. The work is very comprehensive, timely and will be of interest to the field. However, in some places, the data is overinterpreted, and there are a number of awkward statements, and some concerns about the presentation of the work as detailed below:

1. The manuscript would be enhanced with a figure depicting the viral proteins being evaluated.
2. On page 1, line 44, the author suggests that this work would be used to determine virulence of the proteins. The word virulence should be replaced with immunogenicity.
3. The statement on page 1, line 54 in the sentence "...if one protein is attacked by the immune system, weakening the virus, other proteins can continue to assist in its survival." This statement should be revised to reflect the fact that immune targeting of specific proteins is not likely to result in such a response.
4. On page 2, line 2, the author suggests that the presence of unique epitopes in SARS-Cov2 may result in a "...a stronger immune response generated may lead to cytokine storm and immunopathology". While SARS-CoV2 has been shown to induce cytokine storm, it is not likely that this is due to the number of unique epitopes.
5. On page 3, line 6, the author states "...vaccination is indispensable in order to cure an entire population." Vaccines generally do not cure infections, but prevent infection or symptoms. This statement should be corrected.
6. On page 3, line 48, the author states "As opposed to common perception that membrane and spike proteins confer better immunogenic ability, an interesting perception is found from this study that it may be the opposite case in context of SARS-CoV-2 when studied across populations with different HLA-I supertypes." This statement should be qualified to indicate that antibody responses may preferentially target membrane and spike proteins given their locations on the virus, and that this current analysis is geared towards T cell epitopes.
7. The author refers "promiscuous" (e.g. page 5 line 48). The parameters used to determine promiscuous should be stated.
8. On page 6, line 25, in the sentence "...lending support to the theory that these selected epitopes may mount a high immune response in vitro.", the reference to "in vitro" should be "in vivo".
9. On page 13, line 9, the sentence " Another immunogenicity prediction tool, Tcell, was used to predict immunogenic epitopes across two alleles DRB1*01:01 and DRB1*15:01 as it uses PDB files for TCR and there was no structure for other alleles in PDB." What does the author mean by "...PDB files for TCR and there was no structure for other alleles in PDB".
10. The headings and text in the tables, particularly Table 2, need to better described as it's a bit confusing. Particularly, the heading, "clustering with HLA-II epitopes" and "conservation in MSA".

11. The title of Table 2 is "Immunogenic CTL epitopes..." It is not clear if this represents peptides identified as HLA II binders that also bind to HLA I.
12. The author refers to yellow highlights in Table 3, but there is no such yellow highlighting.
13. There is a section on "B cell epitopes", however, this work does not analyze for B cell epitopes. This section should be removed.
14. On page 19, line 51, the author states that "Including B cell epitopes in the vaccination strategy with T cell epitopes may not be a good strategy, and may even be counter-productive." This statement should be revised to reflect that fact that both B and T cell epitopes are important since antibody responses are also critical to generate virus neutralizing antibodies.
15. On page 20, line 9, the author states "This is so done in order to minimize the possible immune system backfiring (22, 23) due to the presence of too many overlapping as well as non-overlapping epitopes in multi-subunit vaccines." It is not clear what is meant by "immune system backfiring".
16. On page 20, line 37, the sentence "Even as this study is important in pointing out the possible mechanisms in contagious nature of SARSCoV-2 , more evidence is required in the form of experiments." It's not clear how the analysis performed in this work point to possible mechanisms of contagious nature of SARS-COV2. This sentence should be revised.

Author's Response to Decision Letter for (RSOS-201141.R0)

See Appendix A.

Decision letter (RSOS-201141.R1)

Dear Dr Mishra,

It is a pleasure to accept your manuscript entitled "Designing of cytotoxic and helper T cell epitope map provides insights into the highly contagious nature of the pandemic novel coronavirus SARS-CoV-2" in its current form for publication in Royal Society Open Science.

COVID-19 rapid publication process:

We are taking steps to expedite the publication of research relevant to the pandemic. If you wish, you can opt to have your paper published as soon as it is ready, rather than waiting for it to be published the scheduled Wednesday.

This means your paper will not be included in the weekly media round-up which the Society sends to journalists ahead of publication. However, it will still appear in the COVID-19 Publishing Collection which journalists will be directed to each week (<https://royalsocietypublishing.org/topic/special-collections/novel-coronavirus-outbreak>).

If you wish to have your paper considered for immediate publication, or to discuss further, please notify openscience_proofs@royalsociety.org and press@royalsociety.org when you respond to this email.

Please ensure that you send to the editorial office an editable version of your accepted manuscript, and individual files for each figure and table included in your manuscript. You can

send these in a zip folder if more convenient. Failure to provide these files may delay the processing of your proof. You may disregard this request if you have already provided these files to the editorial office.

on behalf of Dr John Dalton (Associate Editor) and Steve Brown (Subject Editor)
openscience@royalsociety.org

Appendix A

Associate Editor's comments (Dr John Dalton):

Associate Editor: 1

Comments to the Author:

The paper received a favourable review. However, major revision and additional verification of the statistics applied is asked by the reviewer. The reviewer also felt that the data in some places has been over-interpreted.

Author: Thank you for the favorable review.

Comments to Author:

Reviewers' Comments to Author:

Reviewer: 1

Comments to the Author(s)

In this work, Mishra explore the immunogenicity of thee produced proteins in the SARS-CoV2 RNA genome. Using 2 different computational approaches that takes into account the stages of antigen processing and presentation, MHC I and MHCII binding, and immunogenicity, the authors proposes a list of peptides in almost all of the predicted SARS-CoV2 proteins that may be tested for possible vaccine candidates. The work is very comprehensive, timely and will be of interest to the field. However, in some places, the data is overinterpreted, and there are a number of awkward statements, and some concerns about the presentation of the work as detailed below:

Author: Thank you for your comments that the work is very comprehensive and timely.

1. The manuscript would be enhanced with a figure depicting the viral proteins being evaluated.

Author: This has been done. Thank you.

2. On page 1, line 44, the author suggests that this work would be used to determine virulence of the proteins. The word virulence should be replaced with immunogenicity.

Author: This has been done. Thank you.

3. The statement on page 1, line 54 in the sentence "...if one protein is attacked by the immune system, weakening the virus, other proteins can continue to assist in its survival." This statement should be revised to reflect the fact that immune targeting of specific proteins is not likely to result in such a response.

Author: Thank you, your comments have made the sentence more clear.

4. On page 2, line 2, the author suggests that the presence of unique epitopes in SARS-Cov2 may result in a "...a stronger immune response generated may lead to cytokine storm and immunopathology". While SARS-CoV2 has been shown to induce cytokine storm, it is not likely that this is due to the number of unique epitopes.

Author: The understanding here is that since the human body may be fighting the virus using multiple targets, with T cells being presented with a plethora of epitopes, all of these unique epitopes may be contributing in one way or the other in activating the T cells thereby inducing the cytokine storm. In a latest study, Weiskopf D, Schmitz KS, Raadsen MP, et al. (2020) *Science Immunology* 5,48, DOI: 10.1126/sciimmunol.abd2071, this has also been surmised and found in a preliminary work, and proposed to be carried out in a larger cohort further.

This reference has been included in the main text. And in page 2, line 2 written more clearly as follows: Due to the presence of these unique epitopes in all SARS-CoV-2 proteins, a stronger immune response generated by T cell hyperactivation may lead to cytokine storm and immunopathology and consequently, remote chances of human survival.

5. On page 3, line 6, the author states "...vaccination is indispensable in order to cure an entire population." Vaccines generally do not cure infections, but prevent infection or symptoms. This statement should be corrected.

Author: You are right in stating that vaccines prevent infection. However, studies have pointed out that vaccines can be prophylactic or therapeutic even in viral infections. A therapeutic viral vaccine is also plausible, e.g., anti-HBV therapeutic vaccines. This statement is, hence, modified to include "...prevent infections or cure...".

6. On page 3, line 48, the author states "As opposed to common perception that membrane and spike proteins confer better immunogenic ability, an interesting perception is found from this study that it may be the opposite case in context of SARS-CoV-2 when studied across populations with different HLA-I supertypes." This statement should be qualified to indicate that antibody responses may preferentially target membrane and spike proteins given their locations on the virus, and that this current analysis is geared towards T cell epitopes.

Author: Thank you, this has been included.

7. The author refers “promiscuous” (e.g. page 5 line 48). The parameters used to determine promiscuous should be stated.

Author: This has been stated in the beginning of the main text.

8. On page 6, line 25, in the sentence “...lending support to the theory that these selected epitopes may mount a high immune response *in vitro*.”, the reference to “*in vitro*” should be “*in vivo*”.

Author: These epitopes were selected using *in silico* tools, and need to be validated using *in vitro* assays and *in vivo* also. Included “*in vivo*” in the sentence.

9. On page 13, line 9, the sentence “ Another immunogenicity prediction tool, Tcell, was used to predict immunogenic epitopes across two alleles DRB1*01:01 and DRB1*15:01 as it uses PDB files for TCR and there was no structure for other alleles in PDB.” What does the author mean by “..PDB files for TCR and there was no structure for other alleles in PDB”.

Author: This means that since ITcell tool uses PDB files for TCR as input, and in PDB, the TCR structures corresponding to only these two alleles were present. The statement is rewritten more clearly.

10. The headings and text in the tables, particularly Table 2, need to better described as it's a bit confusing. Particularly, the heading, “clustering with HLA-II epitopes” and “conservation in MSA”.

Author: Changed to “HLA-I epitopes clustered with HLA-II epitopes” and “residue conservation in MSA”.

11. The title of Table 2 is “Immunogenic CTL epitopes...” It is not clear if this represents peptides identified as HLA II binders that also bind to HLA I.

Author: This means that these HLA-I epitopes are also present as a part of longer HLA-II epitope sequence, i.e., clustered.

12. The author refers to yellow highlights in Table 3, but there is no such yellow highlighting.

Author: Highlighting is present, can be changed to lime green.

13. There is a section on “B cell epitopes”, however, this work does not analyze for B cell epitopes. This section should be removed.

Author: B cell epitopes are an important component of adaptive immune response. However, since there is ADE in SARS-CoV-2 infection, this para is needed to strengthen the understanding that focusing on epitopes for CD4+ T cells which can also stimulate antibody responses may be a better option to prevent ADE and at the same time, develop some antibody-based immunity. This is already included in this statement, “Including B cell epitopes in the vaccination strategy with T cell epitopes may not be a good strategy, and may even be counter-productive”. In view of this, this section needs to be kept as such. To reflect it properly, the section title is changed to “ADE and B-cell epitopes”.

14. On page 19, line 51, the author states that “Including B cell epitopes in the vaccination strategy with T cell epitopes may not be a good strategy, and may even be counter-productive.” This statement should be revised to reflect that fact that both B and T cell epitopes are important since antibody responses are also critical to generate virus neutralizing antibodies.

Author: This has already been pointed in the main text: “Even though neutralizing antibody levels are found to be low in Covid19 patients (28, 29), it is expected that CD4+ T cell expansion responses may increase the neutralizing antibody levels (30) and hence quantifying CD4+T cell responses using IFN-gamma ELISPOT assays will be useful”.

15. On page 20, line 9, the author states “This is so done in order to minimize the possible immune system backfiring (22, 23) due to the presence of too many overlapping as well as non-overlapping epitopes in multi-subunit vaccines.” It is not clear what is meant by “immune system backfiring”.

Author: Immune system backfiring is coined for situations when immune system, instead of being protective, leads to immunopathologies.

16. On page 20, line 37, the sentence “Even as this study is important in pointing out the possible mechanisms in contagious nature of SARSCoV-2 , more evidence is required in the form of experiments.” It’s not clear how the analysis performed in this work point to possible mechanisms of contagious nature of SARS-COV2. This sentence should be revised.

Author: It is clarified as follows: “...the possible mechanisms such as immunopathologies arising due to possible T cell hyperactivation, contributing to the contagious nature of SARS-CoV-2...”